# Assessing the Suitability of DME for Powering SI Engines by Analyzing In-Cylinder Pressure Change

**DOI:** 10.3390/s22124505

**Published:** 2022-06-14

**Authors:** Paweł Fabiś

**Affiliations:** Department of Transport and Aviation Engineering, Silesian University of Technology, 44-100 Gliwice, Poland; pawel.fabis@polsl.pl

**Keywords:** internal combustion engine, indicator diagram, dynamics, DME, alternative fuel, in-cylinder pressure

## Abstract

This article discusses an analysis of in-cylinder pressure change during combustion of LPG-DME fuel in IC engines. The aim of the study is to present a method for assessing the possibility of using DME as a combustion activator, and to establish its impact on the process. The study proposes a method for assessing the shift of the maximum value of cylinder pressure as a parameter which enables the impact of DME on the combustion process to be evaluated. The method was developed on the basis of bench tests carried out on an SI engine with a capacity of 1.6 dm^3^.

## 1. Introduction

Recent years have seen extensive progress in the electrification of automotive drive systems. This trend is understandable given the environmental goal to reduce CO_2_ emissions, and the growing use of renewable fuels such as DME and biomethane, to power generators. By coupling electricity generators with internal combustion engines powered by renewable fuels, one can increase energy production without causing negative environmental impact. Using biomass-derived renewable fuels can even reduce the carbon footprint of such energy sources. Therefore, it seems reasonable to conduct research aimed at determining the suitability of selected renewable fuels for powering such internal combustion engines.

One of the known alternative fuels considered as a second-generation renewable fuel is dimethyl ether (DME). Given the extensive options for obtaining this fuel, it may be possible to increase general interest in this energy source and the potential for diversification of conventional fossil fuels. DME can be obtained by recycling waste from the wood and pulp industries. There are other sources from which DME can be sourced, but if they were utilized, the fuel would lose its renewable status.

### 1.1. Fuel Properties

Engine fuels currently in use are mixtures of hydrocarbons characterized by a very wide range of boiling points. Table 1 provides information on the physicochemical parameters of fossil fuels and alternative fuels. A characteristic property of these fuels, that must be taken into consideration, is a low boiling point, especially when compared to methane and LPG. In a free state, these fuels exist as gases, and under certain conditions (e.g., overpressure or low temperature), it is possible to condense them, which increases the density of the energy stored. Both methane (CH_4_) and LPG are commonly used in various European countries to power internal combustion engines. Their physicochemical properties, especially their explosive limits, are similar to those of hydrogen and dimethyl ether (DME). Methane and LPG are often used to power spark ignition engines, while hydrogen is rarely used as a fuel in internal combustion engines, although this option is currently being researched by the Toyota Corporation. The properties of DME make this fuel suitable for powering CI engines [1].

Because the physicochemical properties of DME are similar to liquefied gas, DME can be used as an additive to LPG. It is also possible to use such a mixture as a gaseous fuel for domestic appliances, which is the case, for instance, in regions such as the Middle East (Iran) and Asia (Korea, Japan, and China). In many sectors of the economy (both in industry and institutions), the LPG-DME mixture can be used as a fuel to power engines [3,4], coupled with an electricity generator, or as an energy carrier in heat pumps. The use of the LPG-DME mixture will make it possible, to some extent, to achieve independence from fossil fuels and to reduce their extraction. By that means, energy security will increase and the impact of the unstable prices of fossil energy carriers on national economies will be curbed [1,2].

### 1.2. State of Knowledge

DME fuel was often addressed in the literature as early as the end of the 20th century. There were high hopes regarding this energy carrier, but unfortunately that did not translate into large-scale utilization of the fuel for powering internal combustion engines. However, the concept of using DME as an engine fuel is gaining popularity. Selected studies on the problems of fuel combustion and associated emissions are discussed below.

DME, like hydrogen, can be used as an activator in the combustion process, affecting its course. A higher rate of hydrogen and DME combustion accelerates the natural gas combustion initiation process, and should also have an effect on LPG combustion. Both these gaseous fuels are currently used extensively to power motor vehicles.

There are studies [5,6,7,8,9,10,11] which address the possibilities for using DME as a fuel to power internal combustion engines. They present the current state of knowledge on the use of renewable and alternative fuels with regard to the problem of internal combustion engine powering in vehicles.

In a study by Kwak J., et al., road tests of emissions in ppm were carried out for four vehicles [5]. Two of the vehicles were powered by CI engines, the other two by SI engines. Four types of fuel were included in the assessment: diesel fuel, LPG, CNG, and DME. Each of the vehicles tested was powered by a different fuel, and the speeds at which exhaust gases were sampled were as follows: 50 km/h, 80 km/h, and 100 km/h. The size of the nanoparticles emitted, and their quantity as a function of vehicle speed, were adopted as the basic emission parameter. The study concluded that the size of the nanoparticles emitted was largest in the case of DME. Moreover, the nucleation mode particle concentration in the DME feed was proportional to the NO_x_ emission.

Lee S., Oh S., Choi Y., and Kang K. attempted to determine the emissions from a 2.7 dm^3^ engine fueled with a mixture of LPG and DME [6]. The fuel was fed indirectly to the intake manifold in a liquid phase. The parameters measured were torque, exhaust gas temperature, and combustion process stability. Furthermore, the emission of basic exhaust gas components (CO, THC, and NO_x_) was also verified at the rotational speed of 1800 rpm. The torque values measured for the throttle wide open were recorded within the rotational speed range of 1800–5200 rpm. The highest torque values within the entire range of rotational speeds were obtained for the LPG fuel (70% butane and 30% propane), while the lowest value was recorded for n-butane with a 20% fraction of DME. Exhaust gas temperature measurements revealed the highest temperature values for the n-butane and DME mixture fuel. The temperature difference between the highest and the lowest point was approximately 80 °C, a relatively large value. The authors also determined the effect of the DME share in the mixture on the combustion process. Their research made it possible to determine the dynamics of the combustion process by establishing the pressure increases in the cylinder. Their results showed that the addition of DME accelerated the combustion process. Moreover, the authors found that once the excess air ratio was modified it was possible to use the DME additive without any further modifications.

The impact of a DME additive on LPG was assessed by Pathak S.K., Sood V., Singh Y., Gupta S., and Channiwala S.A. [7]. The authors examined the behavior of a vehicle powered by an SI engine fed with LPG-DME fuel against the EURO IV exhaust gas purity standard. The engine of the test vehicle was equipped with an adaptive gas fueling system, where the gas was delivered to the engine in the indirect evaporation phase (to the intake manifold). The test program included an assessment of the changes to the exhaust gas emissions against various fuels (gasoline, LPG, and an 80/20% mixture of LPG and DME). The researchers were mainly focused on the high NO_x_ emissions attributable to the DME fuel. The reason for its increased emission of nitrogen oxides was the higher temperature in the combustion process due to a slight depletion of the mixture, and the factor responsible for the mixture depletion was the small amount of oxygen carried by DME. The study also presented the results of an analysis of carbon dioxide emissions and fuel consumption. Attention was paid to a slightly higher fuel consumption with DME, which was caused by the fuel’s lower energy value. The CO_2_ emission results discussed in the study confirm the possibility of reducing the volume of particles introduced into the environment. Carbon dioxide emissions from a fuel containing 20% of DME are lower by approximately 10% than the emissions attributable to gasoline. The authors advocated using the DME additive with LPG, without the need to make any technical modifications to the injection system.

Other studies [12,13,14,15] of LPG-DME fuels have focused on determination of the effect of the ignition advance angle, the composition of the fuel-air mixture on the operational parameters of engines, and analysis of the potential for knocking combustion. A study by Chen Y., Zhang Q., Li M., Yuan M., Wu D., Qian X. [14] presented the results of research on wave propagation in the LPG-DME fuel combustion process. The authors established the velocity and value of the shock wave pressure in a closed tank filled with fuel comprising an LPG and DME mixture.

An important aspect of this problem was describing and testing the process of the LPG and DME mixture combustion. Knowledge of the combustion process would make it possible to adapt engines to meet the combustion requirements of a specific fuel [16,17,18]. The authors of the study attempted to determine the time delay in the initiation of ignition of a stoichiometric mixture of methane and DME. Their tests involved using a pipe in which the shockwave propagation velocity resulting from the ignition of gaseous fuels was controlled. The test results thus obtained were compared with data retrieved from the literature on the subject. The study also addressed the possibility of using DME to power HCCI, SEHCCI or PCCI engines. Computer calculations were carried out to estimate engine operation indicators in the event that DME fuel was used to power the engine.

DME is not the only fuel currently being considered to power IC engines. Some studies [19,20,21,22] have discussed the possibilities of using renewable plant-derived fuels. These fuels were fed into diesel engines, where the combustion process and the potential for changing the EGR valve settings were examined. Other studies have discussed the possibility of using ethanol to power IC engines. Manufacturers offering vehicles with engines adapted for ethanol fuel are featured in another study, which also analyses options for running on CNG, biodiesel, and electricity [21]. The most recent study [13,23,24,25] presents numerical simulations of a four-cylinder IC engine simulated in an AVL Boost environment. These simulations made it possible to establish the relevant CO, HC, and NOx emissions, as well as to determine engine operating parameters such as power, cylinder pressure, and temperature in the combustion chamber. LPG, ethanol, and gasoline were used as fuels in this case.

In light of the above, efforts to establish if DME is suitable for powering SI engines seem justified.

## 2. Research Objects and Measurement Set-Up

Assuming that DME is used as an additive, we must consider how this ingredient is to be introduced into LPG. Application of the DME additive may be considered, for example, at the LPG bottling stage, at a plant where it is possible to introduce the assumed dose of ether, either during or before charging. It is also possible to develop a suitable mixing system and deploy it at a refueling station. A diagram of such an installation, designed and built at the Faculty of Transport and Aviation Engineering of the Silesian University of Technology, is shown in Figure 1.

The fuel supplied to the test vehicle’s engine was produced by means of the apparatus shown above. The research in question has made it possible to assess the combustion efficiency of the mixture of LPG and DME fuel.

The system demonstrated in Figure 2 enables production of any mixtures with different proportions of DME and LPG. Determination of the DME fraction in the mixture of the test fuel makes it possible to identify its effect on the course of the combustion process. The designed research program envisaged experiments performed for the following fuel types, differing in terms of the mass fraction of DME:7% DME, 93% LPG,11% DME, 89% LPG,14% DME, 86% LPG,17% DME, 83% LPG,21% DME, 79% LPG,26% DME, 74% LPG,30% DME, 70% LPG,100% LPG (40-to-60 propane/butane ratio mixtures).

The mixtures prepared in this manner were supplied to the test vehicle’s engine via an adequately configured, additional supply system (Figure 2). The tests were carried out for six engine loads of 21%, 33%, 48%, 69%, 90%, and 100%.

The test program was run on an OPEL ASTRA engine(GM manufacturer, Flint, MI, USA). The gaseous fuel supply system of this engine was configured so as to enable the fuel to be introduced indirectly (into the intake manifold channels) in the vapor phase. The most important parameters of the test engine have been provided in Table 2.

The car engine’s operating parameters were determined by analyzing its characteristics, obtained by means of a type FLA 203 Bosch chassis dynamometer, with reference to pre-prepared mixtures with a varying mass fraction of DME. A simplified diagram of the test bench has been provided in Figure 2.

The test bench featured transducers and sensors enabling identification of the engine operating condition. The basic control and measurement systems, ensuring continuous recording of the engine’s operational status, made it possible to measure the following parameters:In-cylinder pressure;Crank angle with TDC;Wheel power;Intake manifold pressure;Exhaust gas temperature;Mass flow of the gaseous fuel delivered to the engine.

The vehicle was tested by means of the BOSCH FLA 203 chassis dynamometer, which enables testing of automotive vehicles and motorcycles with single-axle drive systems. The device makes it possible to define such parameters as:Engine power;Engine torque;Driving force;Power on the wheel;Acceleration.

Additionally, using the dynamometer, it is possible to verify the correctness of indications of tachometric devices: speedometers and odometers. The chassis dynamometer (Figure 3) measures the force transmitted from the vehicle wheels to the dynamometer rollers, as well as rolling resistance. Table 3 contains the basic technical specifications of the dynamometer in question.

Before proceeding with the measurements, the test vehicle was prepared in accordance with the guidelines contained in the dynamometer’s operating manual [27]. The vehicle was placed on the dynamometer rollers, so that the vehicle’s axis of symmetry was perpendicular to the axis of symmetry of the rollers, and then the pressure in the tires was set correctly. The vehicle was then tested. The dynamic power was measured (the engine’s external characteristics were established) by accelerating the car on the gear corresponding to the direct ratio until the maximum engine revolutions were obtained. The dynamometer’s computer measured the driving force transmitted from the vehicle wheels to the dynamometer rollers.

A force sensor was placed on a known arm, which made it possible to calculate the torque value. The power was calculated empirically by converting the torque value. After reaching the maximum engine speed, the drive was disengaged. The dynamometer rollers spun due to their inertia, propelling the vehicle’s wheels and the driveline. In this way, the resistances related to the resistance emerging at the dynamometer rollers and the resistance of the drive system of the test vehicle were established. Based on this information, the computer was able to determine the engine’s external characteristics, comprising the corrected characteristic, because the results of the power measurement were supplemented with the power of the dynamometer and the power attributable to the resistances of the test vehicle’s drive system.

In order to determine parameters such as compression pressure or ignition timing, it was necessary first of all to establish the crankshaft position and its rotational speed in an accurate manner. To this end, the test vehicle was equipped with the KISTLER 2613B crankshaft position marker (Figure 4). The sensor’s small dimensions made it possible to conduct the tests in a very wide range, while maintaining highly accurate measurements.

The position sensor was mounted on the crankshaft pulley (Figure 5) of the engine subject to tests. Then the sensor was precisely set so that the upper piston position indicated by the marker coincided with the actual TDC of the engine piston. The TDC of the piston was verified by observing the maximum in-cylinder pressure as measured by the gauge system.

The marker’s signal resolution was 0.1–6°, and its dynamic accuracy at 10,000 rpm was +0.02°. The speed range at which the marker could be used was set at 1–20,000 rpm.

The in-cylinder pressure was measured using a system comprising a type 6121 piezoelectric quartz pressure sensor and a type 5011 charge amplifier (Figure 5). Both devices were KISTLER branded products [28]. The parameters were recorded by a PC connected to the measuring system via a data acquisition card (Table 4).

Such a pressure measurement system was resistant to the disturbances attributable to high temperatures and dynamic loads. A pressure transducer was fixed at the cylinder head measuring the pressure changes in the fourth cylinder of the engine of the Opel Astra subject to the tests. The pressure in the cylinder could be measured once a hole had been drilled in the cylinder head up to the engine combustion chamber. The transducer in use was a device suitable for pressure measurements under particularly difficult conditions, such as those in the engine combustion chamber. The transmitter’s operating range was 0–250 bar. The temperature range in which such a converter could operate without the need for additional cooling was specified at 20–350 °C.

The charge amplifier was an independent device that enabled conversion of a capacitive signal [pC] into a proportional voltage signal [V]. The device featured control buttons located on the front panel, making it possible to select the measurement range and to change the time base. The amplifier also comprised signal filtering circuits. The technical specifications of the charge amplifier are provided in Table 3.

A PCI 6143 S series data acquisition card, manufactured by National Instruments (Austin, TX, USA) was used to collect data from all sensors and transducers.

The card featured 8 different 16-bit channels with a maximum sampling frequency of 100 kHz. The NI-DAQ PCI acquisition card generated a code which enabled observation of the signals acquired in the LabView program or in other card handling programs. The device was an integral part of the computer architecture, while the data acquisition and flow algorithms did not change when transferred via the recording system. The last feature was particularly important from the point of view of measurement errors attributable to the measurement system.

## 3. Results and Discussion

The results of the bench tests, in which a large number of engine operation cycles were recorded, were analyzed by determining the mean indicated pressure value. This was done by determining the mean indicated pressure for all courses of combustion pressure and by calculating the arithmetic mean (Indicated Mean Effective Pressure—*imep*) [29,30].

The area enclosed by the curves in the indicator corresponded to the indicated work affecting the piston crown. The indicated pressure and *imep* were calculated as indicated work divided by the stroke volume of the cylinder. The *imep* value was independent of the size and number of cylinders and the rpm value. The actual *imep* calculations were carried out for 200 cycles of engine operation. The *imep* parameter was defined as follows:(1)$$imep=\frac{W_{i}}{V_{s}}$$
where:*W_i_*—indicated work [Nm],*V_s_*—cylinder capacity [m^3^].

In real time, *imep* was calculated using the following equation:(2)$$imep=\sum_{i=0}^{719}p_{i}\frac{dV}{V}$$

For the mean indicated pressures thus determined, the arithmetic mean was calculated according to the following formula:(3)$$\bar{imep}=\frac{\sum_{i=0}^{N}imep}{N}$$
where:*N*—cycle number.

The indicated pressure values for the DME share values envisaged in the tests at a selected load of 100% are given in Figure 6.

The test results thus obtained were analyzed in order to assess their quality with reference to the calculations. This analysis made it possible to eliminate those engine operating cycles in which combustion process anomalies were observed. The reasons why individual engine operating cycles were eliminated were knocking combustion and misfires.

For the established mean indicated pressure, the value of the coefficient of covariance was determined according to the following relationship:(4)$$COV\_imep=\frac{\sqrt{\frac{\sum {\left(p_{i}-\bar{p_{i}}\right)}^{2}}{N-1}}}{\bar{p_{i}}}$$
where:*N*—number of cycles,${\bar{p}}_{i}$—mean value of indicated mean pressure.

Sample values of dispersion of mean indicated pressure (*COV_imep*) are provided in Figure 7.

Inferring from the results presented in Figure 7, no value of mean indicated pressure exceeded the limit of 5%, which clearly indicates that the equipment in use was operating correctly, and makes it possible to run further calculations aimed at assessing the suitability of LPG-DME mixtures for vehicle powering.

This study’s analysis, of the possibility of using the LPG-DME mixture, is based on the determination of the impact of the DME proportion on the shift of the maximum value of in-cylinder pressure. Many publications refer to the specific property of DME used as a fuel which activates the combustion process. Based on preliminary analyses preceding the publication of this study, it can be assumed that the addition of DME to LPG causes changes in the course of the combustion process, and more specifically, triggers a change in the angle of maximum pressure in the cylinder (Figure 8).

Having analyzed the graph in Figure 8, it can be concluded that the maximum value of in-cylinder pressure shifts with respect to TDC. Assuming LPG as the base fuel, a trend can be observed whereby the maximum pressure is first delayed for consecutive DME fractions in the fuel and, in the case of maximum values, this shift decreases again. This means that the trend is not proportional to the DME proportion in the mixture. This is due to a change in the combustion process. Small fractions of DME accelerate the charge burnout. With medium DME fractions, the changes are more pronounced, and in the case of the maximum DME fractions tested, the changes dwindle again. This can be explained by the large amount of fuel with a low octane number and a relatively large amount of oxygen (contained in the DME molecule). A depleted mixture burns much slower, and therefore change in the position of the maximum pressure value is smaller.

An attempt was also made to determine the impact of the correction of the ignition advance angle on the shift of the maximum value of in-cylinder pressure. The changes in the position of maximum in-cylinder pressure were visualized using simple statistical functions. The maximum in-cylinder pressure values were determined first, followed by the angle of the crankshaft rotation at which the maximum pressure value was observed. Then, a median of the determined angle values was calculated, the results of which are presented in the diagrams below (Figure 9).

The above graphs imply a tendency for the maximum in-cylinder pressure to shift relative to the TDC of the piston. The moment at which the maximum pressure value appears is delayed as the DME content increases. This trend is independent of the set engine load, the ignition timing angle correction, and the rotational speed. This confirms that DME actively changes the course of the combustion process.

SI engine fueling with the LPG-DME mixture is characterized by different parameters compared to the base fuel (LPG). In this study, an attempt was made to develop a method for analyzing such engine behavior by assuming the angle at which the maximum in-cylinder pressure occurred as a parameter describing these changes. The research results were analyzed, and the changes in the angle of the maximum in-cylinder pressure were determined. In conclusion, as the amount of DME changed in the mixture, so did the maximum pressure angle.

## 4. Summary

Workplace data acquisition involved recording all processes, including those that might contain erroneous information or quality parameters of processes that did not fall within generally accepted ranges. It was therefore necessary to perform a pre-selection which allowed for any false results to be rejected. This elimination was possible using mean indicated pressure. Analysis of the mean indicated pressure value, and of its covariance above all, made it possible to reject results which diverged from the others. This, in turn, allowed for such data to be adopted for further analysis and included in a limited field of rational values. No selected cylinder pressure courses contained anomalies, and according to the pre-determined coefficient of covariance, the maximum in-cylinder pressure deviation did not exceed 5%.

The fact that the timing advance angle was corrected during the measurements (0°, +3° and +6°) meant that the mean indicated pressure (*imep*) values changed to a small extent. The higher the correction value, the higher the *imep* value. The mean indicated pressure values thus obtained for the assumed timing advance angle corrections were consistent with the relevant literature data. Increasing the correction further above +6° did not bring the expected results. The dynamic of the changes in the indicated pressure value actually declined, which implies that increasing the correction factor is pointless.

Changes in the angle of peak pressure nearly always delayed its occurrence. This meant that the greater the proportion of DME in the mixture, the later the maximum in-cylinder pressure appeared. This phenomenon was observed regardless of the engine speed, its load, and the introduced correction of the ignition advance angle (Figure 10 and Figure 11).

The ignition advance angle correction affected, within certain limits, the shift of the maximum pressure value. The greatest changes, in this respect, applied to the correction of +3° CA. Further increasing the timing advance correction was pointless, as it had little effect on the changes in the position of the maximum pressure value. This phenomenon was observed for all tested LPG and DME mixtures.

With regard to engine speed, the changes were slightly more evident. As the crankshaft rotational speed increased, a change in the position of the maximum pressure towards larger angles was observed once the piston TDC was attained. This change delayed the time when the maximum pressure value was reached, and triggered a subjective change in the noise emitted by the engine. The subjective change in noise should be understood as the emission of sound as perceived by the environment. The noise level was not measured in any way, since it was a subjective feeling of those who performed the tests. This subject should be verified in subsequent research projects.

The main subject of this study was the analysis of engine in-cylinder parameters. The analysis of the emission levels of toxic exhaust gas components was only an addition, allowing for a more solid confirmation of DME potential as an SI engine fuel. Emission measurements were carried out only for few selected points featuring engine steady operating conditions. The next step, after selecting the most promising fuel component composition (LPG-DME), will be an in-depth analysis of exhaust gas components and the determination of emissions for selected driving tests.

## Figures and Tables

**Figure 1 sensors-22-04505-f001:**
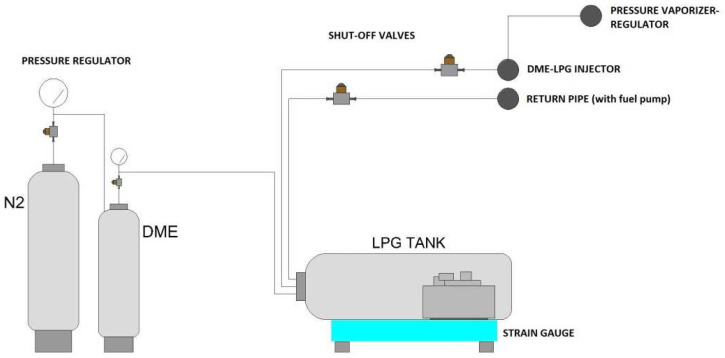
LPG-DME mixture preparation system [25].

**Figure 2 sensors-22-04505-f002:**
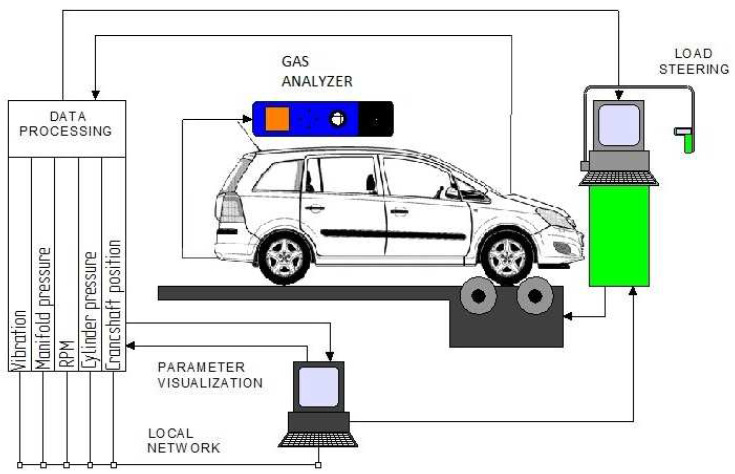
Test bench diagram [26].

**Figure 3 sensors-22-04505-f003:**
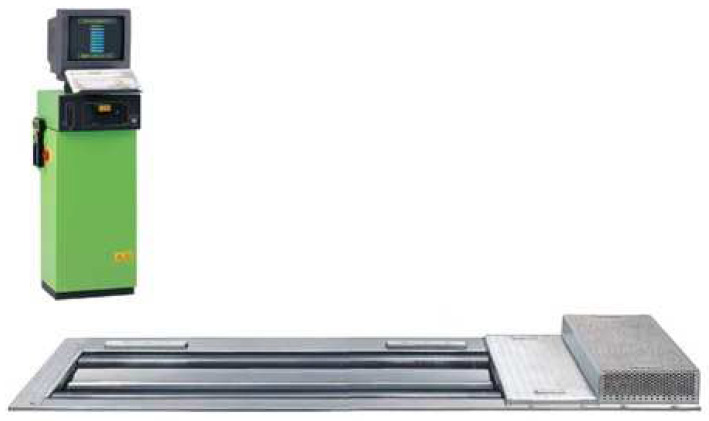
BOSCH FLA203 dynamometer stand [27].

**Figure 4 sensors-22-04505-f004:**
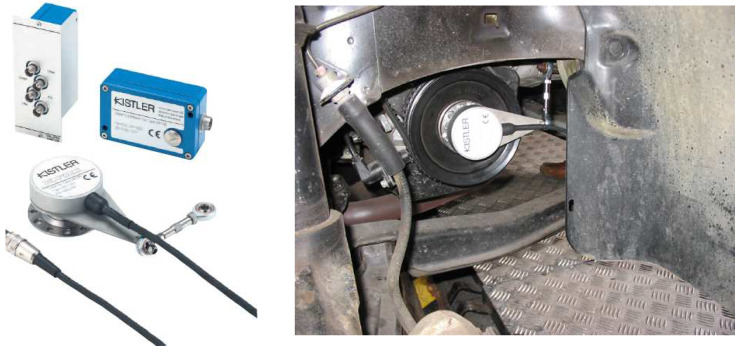
Crankshaft position sensor [28].

**Figure 5 sensors-22-04505-f005:**
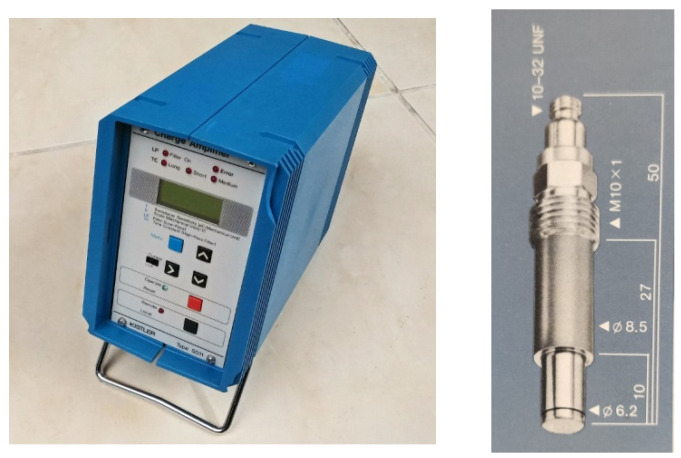
Pressure sensor and signal amplifier [28].

**Figure 6 sensors-22-04505-f006:**
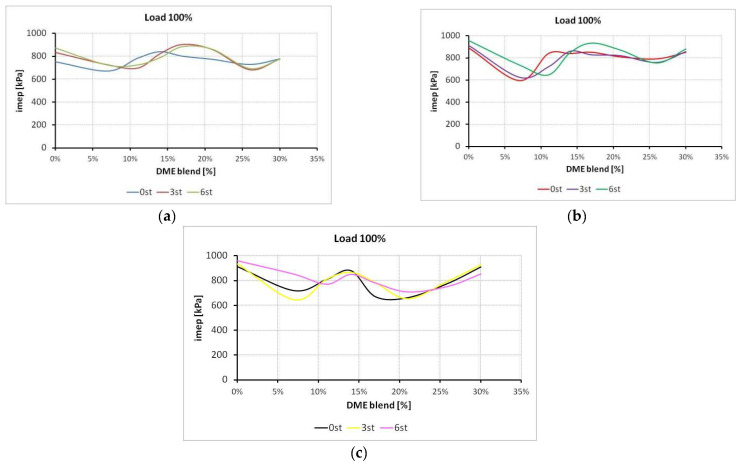
*Imep* value for three speeds: (**a**) 2000 rpm, (**b**) 2500 rpm, and (**c**) 3000 rpm.

**Figure 7 sensors-22-04505-f007:**
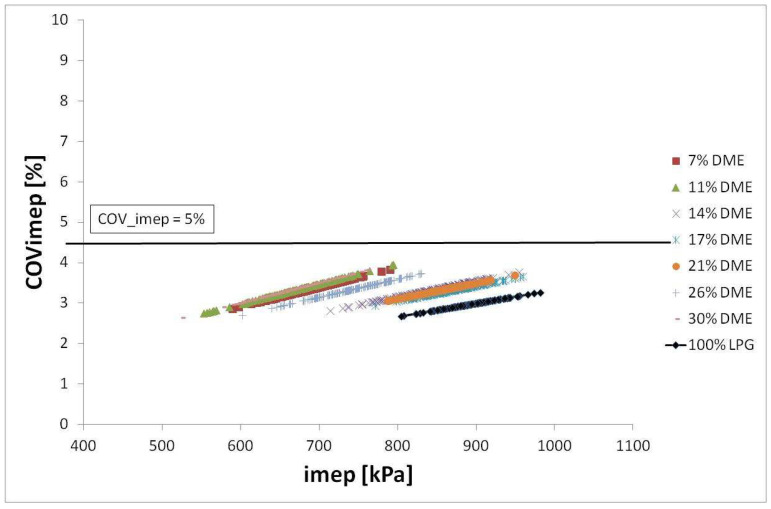
Covariance of *imep* values.

**Figure 8 sensors-22-04505-f008:**
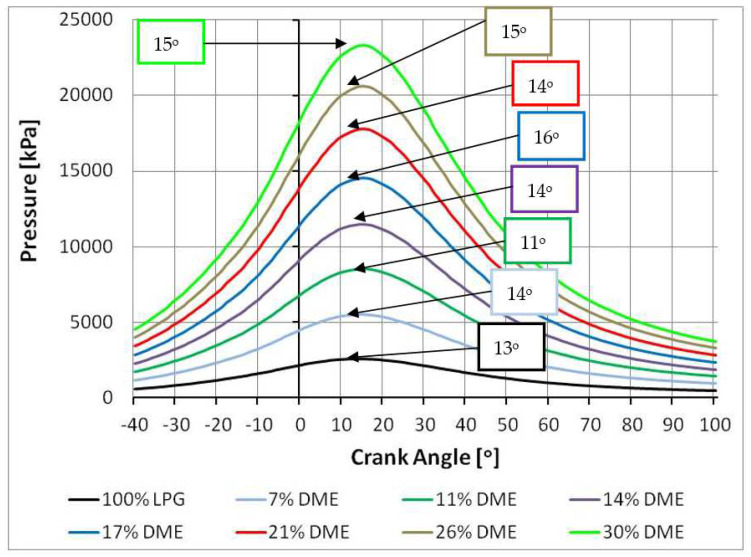
Maximum in-cylinder pressure value vs. crank angle (33% load and 3000 RPM, presented as the sum of the consecutive data series).

**Figure 9 sensors-22-04505-f009:**
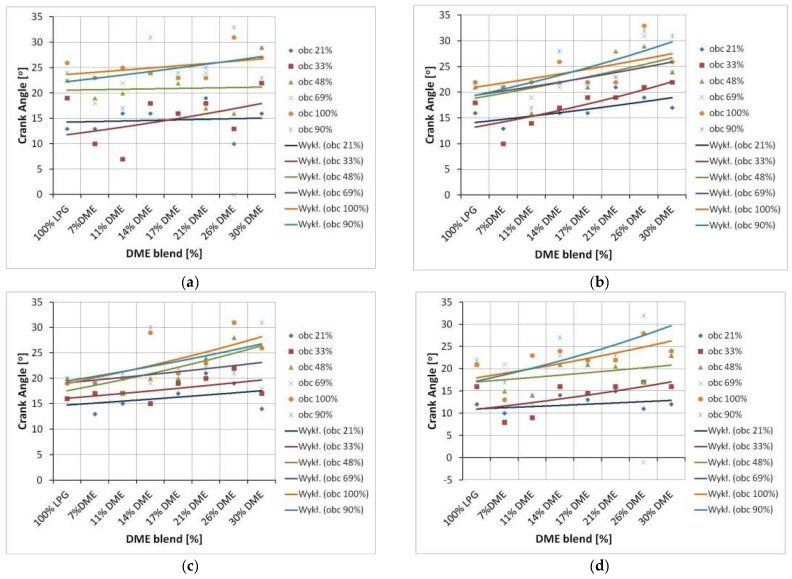
Shift of maximum in-cylinder pressure values relative to TDC. (**a**) 2000 rpm and 0° CA correction, (**b**) 2500 rpm and 0° CA correction, (**c**) 3000 rpm and 0° CA correction, (**d**) 2000 rpm and +3° CA correction, (**e**) 2500 rpm and +3° CA correction, (**f**) 3000 rpm and +3° CA correction, (**g**) 2000 rpm and +6° CA correction, (**h**) 2500 rpm and +6° CA correction, (**i**) 3000 rpm and +6° CA correction.

**Figure 10 sensors-22-04505-f010:**
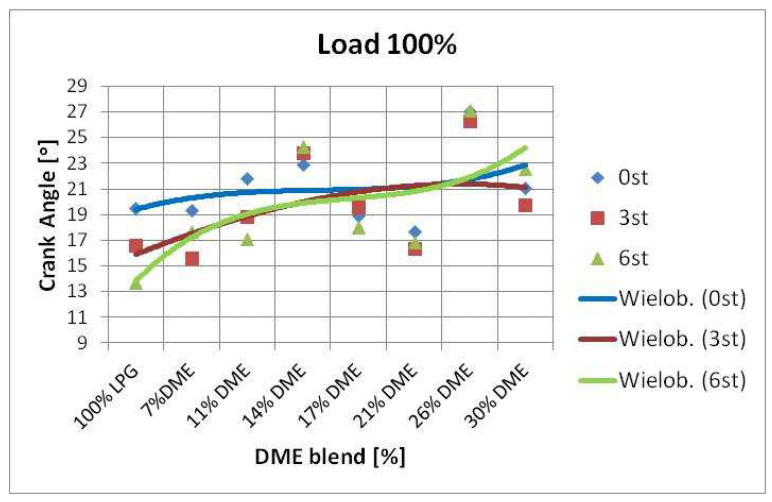
Change of maximum in-cylinder pressure angle vs. RPM.

**Figure 11 sensors-22-04505-f011:**
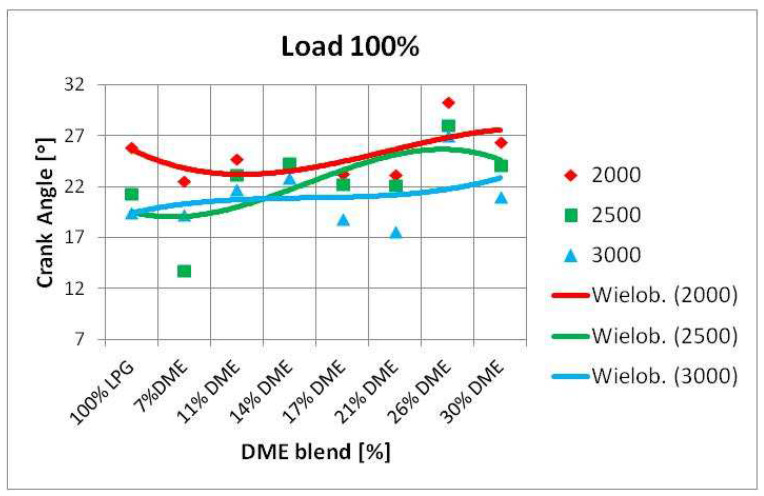
Change of maximum in-cylinder pressure angle vs. crank angle correction.

**Table 1 sensors-22-04505-t001:** Physicochemical properties of chosen fuels [1,2].

Parameter	Methane	LPG	Dimethyl Ether (DME)	Petrol	Diesel
Formula	CH_4_	C_3_H_8_-C_4_H_10_	CH_3_OCH_3_	C_7_H_16_	C_14_H_30_
Molecular weight (g/mol)	16.4	44.1	46.07	100.2	198.4
Density (g/cm^3^)	0.720	0.2	0.661 (liquid) 2.057 (vapor)	0.737	0.856
Boiling temperature (°C)	−162	~22	−24.9	17–220	140–380
Octane number	130	~105		80–100	
Cetane number			55–60		40–55
Lower calorific value (MJ/kg)	50.2	46	28.8	43.47	41.66
Stoichiometric air/fuel ratio (kg/kg)	17.2	15.5	9.0	14.7	14.6
Self-ignition temperature (°C)	540–650	540	350	228–300	150–250
Burning velocity (cm/s)	30–33.8	38	42.9–61	30–60	
Sulfur (ppm)	7–25	10	0	~200	~250

**Table 2 sensors-22-04505-t002:** Characteristics of the test engine.

Number of Cylinders	Inline 4
Power/Rotational speed	55 kW/5200 rpm
Torque/Rotational speed	128 Nm/2800 rpm
Displacement	1598 cm^3^
Bore	79.0 mm
Stroke	81.5 mm
Compression ratio	9.6

**Table 3 sensors-22-04505-t003:** Dynamometer characteristics.

Engine power range	0–300 kW
Measurement accuracy	±0.5%; ±2 kW
Velocity range	0–260 km/h
Velocity measurement accuracy	±1%; ±0.9 km/h
Dynamometer power	25 kW at 20 km/h 150 kW at 130 kmh 260 kW at 260 km/h
Accuracy	±0.5%; ±2 kW
Maximum resistance power	6000 N
Rolling resistance change	0–50 kW
Accuracy	0.1 kW at 90 km/h
Air resistance change	0–50 kW
Accuracy	0.1 kW at 90 km/h
RPM range	0–15,000 min^−1^
RPM accuracy	10 min^−1^

**Table 4 sensors-22-04505-t004:** Amplifier characteristics [28].

Measurement range	±10 ÷ 999,000 pC
Sensor sensitivity	0.1 ÷ 11,000 pC/bar
Voltage signal conversion	0.001 ÷ 9,999,000 bar/V
Output voltage	±10 V
Output current	±0 ÷ 5 mA
Output impedance	10 ± 1% Ω
Constant value: - Long - Medium - Short	undefined 1 ÷ 10,000 s 0.01 ÷ 100 s
Device mass	2 kg

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
