# Peer review of "Assessing the Suitability of DME for Powering SI Engines by Analyzing In-Cylinder Pressure Change"

_sensors, 2022, doi:10.3390/s22124505_

Round 1

Reviewer 1 Report

General comment

The draft presents an interesting topic but it is not well presented and discussed. Important features of a scientific article are missing. The work tastes more like an internal technical report of a company than a scientific article.

In the various parts of the work, there are no meaningful interpretations of the results but only a flat description of the results.

Some punctual notations 

  1. The introduction is absolutely insufficient, it must be completely rewritten. The state of the art is absent, all the references, from the number 8 to the 32nd, are not cited nor commented in the text.
  2. Some graphics are unreadable (see Figure 5):
  3. Figure 1 is not referenced in the text.
  4. Graph 1 is absolutely incomprehensible and the link with the bibliographic reference [1] is not seen.
  5. Figure 6 without the pressure scale.
  6. A legend with the acronyms is needed.
  7. English needs to be revised.
  8. How many cycles (N), the author uses to evaluate the imep? For example, N is the number used in the expression (3).
  9. At row 287:  indicated work instead of indication work.
  10. 10000 RPM instead of 10,000 min-1.

Author Response

Dear Reviewer,

Thank you for the review and I am sending the answer along with the corrected article.

  1. I corrected the introduction to the article. The introduction has been redrafted and the "current state of knowledge" section added. In this part of the study, studies that were not featured in the previous version of the text were added.
  2. All illegible drawings have been corrected.I hope that their current quality will be sufficient.
  3. Chart 1 has been removed from the study. It seems that there is no need to mention this relationship. The new version of the introduction is sufficient and does not need to be supplemented with this study.
  4. As suggested, the graph in figure 6 has been corrected, the scale has been added. Additionally, in the description, the scale value was commented on.
  5. At the end of the study, I added the nomenclature - as suggested.
  6. The article was resent for English corrections.However, a revised version will be sent as soon as possible.
  7. In the case of calculations related to indicated pressure, the minimum number of working cycles (N samples) is 150 cycles.These cycles do not contain combustion abnormalities such as knocking.These are cycles after initial segregation.
  8. On line 287, I corrected the wrong name.
  9. Throughout the study, I changed 10.000 min-1 to 10.000 RPM.

I also supplemented the description of the obtained results to make it a bit more complete. I hope the current version of the article is more appropriate.

Reviewer 2 Report

In the used literature, the empty literature No.33 is mentioned. Please add or remove!

I request that the tables and figures be numbered correctly and incorporated into the text according to the relevant figures and tables.

On line 81 83, the authors mention "emitted particles", which will serve as a basic emission parameter. It is not clear what the particles are and therefore I consider this to be an inaccurate statement. Please clarify and add to the text.

I agree with the author of the manuscript that the method of dosing DME is very important, because the combustion process may differ from the perfection of mixing the mixture.

It would be good to unify the scale on the y-axis more appropriately and it would be visually better if the range 0-1000 was used for all 3 graphs. 

On line 265 there is a reference to "Fig.7.10". It is not clear what image the author refers to. 

In chapter 3 the parameter "imep" is mentioned but on line 283 "mep" is mentioned. Is it an error or another parameter that is not explained? Please modify or add the meaning of the "mep" parameter!

There are abbreviations in the text of the manuscript that are not explained, so it is advisable to add Nomenclature.

It would be good to explain the last sentence on line 385-387, especially "subjective change". A more precise statement is missing here.

Author Response

Dear Reviewer,

Thank you for the review and I am sending the answer along with the corrected article.

  1. I removed the empty item 33 from the literature. Additionally, I looked through the literature again and put it in order.
  2. I corrected the numbering of figures and tables - as suggested.
  3. Lines 81-83, I corrected the particle naming bug and introduced the correct terminology.
  4. The graphs in Figure 6 have been corrected. All axes have been modified and unified - as suggested.
  5. In line 265 I corrected the marking of the drawing.
  6. In line 283 I corrected the description of the "mep" parameter to "imep".
  7. At the end of the study, I added the nomenclature - as suggested.
  8. The term used in lines 385-387 has been clarified.

Round 2

Reviewer 1 Report

The authors, in the introduction, talk a lot about works in which the emissions of engines powered with DME blends were studied in detail. Furthermore, in Figure 2, the overall view of the experimental plant is shown, where the presence of a gas analyzer is seen but, in the article, no measurement and evaluation are shown in this regard.

This is the real lack of the present work to make it complete.

It is advisable to insert a final paragraph in which a detailed emissions analysis appears as future work.

Author Response

As suggested, a paragraph was added about future research devoted to extending the information on the results of emission measurements for fuels selected in terms of DME share